# Berberine Promotes Beige Adipogenic Signatures of 3T3-L1 Cells by Regulating Post-transcriptional Events

**DOI:** 10.3390/cells8060632

**Published:** 2019-06-23

**Authors:** Ying-Chin Lin, Yuan-Chii Lee, Ying-Ju Lin, Jung-Chun Lin

**Affiliations:** 1Department of Family Medicine, Shuang Ho Hospital, Taipei Medical University, New Taipei City 235, Taiwan; greening1990@gmail.com; 2Department of Family Medicine, School of Medicine, College of Medicine, Taipei Medical University, Taipei 110, Taiwan; 3Department of Health Examination, Wan Fang Hospital, Taipei Medical University, Taipei 116, Taiwan; 4Department of Geriatric Medicine, School of Medicine, College of Medicine, Taipei Medical University, Taipei 110, Taiwan; 5Graduate Institute of Biomedical Informatics, Taipei Medical University, Taipei 110, Taiwan; ycgl@tmu.edu.tw; 6School of Chinese Medicine, China Medical University, Taichung 404, Taiwan; yjlin@mail.cmu.edu.tw; 7School of Medical Laboratory Science and Biotechnology, College of Medical Science and Technology, Taipei Medical University, Taipei 110, Taiwan; 8PhD Program in Medicine Biotechnology, College of Medical Science and Technology, Taipei Medical University, Taipei 110, Taiwan

**Keywords:** berberine, beige adipocyte, miR-92a, RBM4a

## Abstract

Induced brown adipocytes (also referred to as beige cells) execute thermogenesis, as do the classical adipocytes by consuming stored lipids, being related to metabolic homeostasis. Treatment of phytochemicals, including berberine (BBR), was reported to induce conversion from white adipocytes to beige cells. In this study, results of microRNA (miRNA)-seq analyses revealed a decrease in *miR-92a*, of which the transcription is driven by the *c13orf25* promoter in BBR-treated 3T3-L1 cells. BBR treatment manipulated the expressions of SP1 and MYC, in turn, reducing the activity of the *c13orf25* promoter. A decrease in *miR-92a* led to an increase in RNA-binding motif protein 4a (RBM4a) expression, which facilitated the beige adipogenesis. Overexpression of *miR-92a* or depletion of RBM4a reversely interfered with the impact of BBR treatment on the beige adipogenic signatures, gene expressions, and splicing events in 3T3-L1 cells. Our findings demonstrated that BBR treatment enhanced beige adipogenesis of 3T3-L1 cells through transcription-coupled post-transcriptional regulation.

## 1. Introduction

Excess energy intake leads to hypertrophy and hyperplasia of white adipocytes (WAs), which results in obesity and related metabolic syndromes [1,2]. In contrast, brown adipocytes (BAs) execute active metabolism and thermogenesis with abundant mitochondria [3]. Classical BAs originate from *Myf5*^+^-mesodermal precursors, and inducible BAs (also referred as beige cells) could be derived within white adipose tissues (WATs) and originate from the *Myf5*^−^lineage [4,5]. Cold exposure, presence of β3-adrenergic receptor (β3-AR) agonists, and nutrient intake drove trans-differentiation of WAs to inducible beige cells, which was proposed a therapeutic intervention to counteract obesity [6,7,8]. Berberine (BBR) is a naturally occurring alkaloid present in *Coptis chinensis* (Chinese goldthread) and *Hydrastis canadensis* (goldenseal), which facilitates adaptive thermogenesis and enhances browning of WAs in rodents [9,10]. In vivo animal models approved the effect of BBR on combating hyperlipidemia and the accumulation of WAs [11]. Although the health benefit of BBR is widely reported, the detailed mechanism, such as transcriptional or post-transcriptional control, involved in BBR-enhanced brown adipogenesis is yet uncharacterized.

Post-transcriptional control, including alternative splicing (AS), microRNA (miRNA)-mediated gene regulation, and mRNA surveillance, constitutes a spatiotemporally mechanism for determining cellular fates and functions [12,13,14]. Y-box binding protein 2 (Ybx2) was demonstrated to enhance the stability of the peroxisome proliferator activated receptor (PPAR) gamma coactivator-1 (PGC-1α) transcripts, which acted as the critical factor for activating the thermogenic program of BATs [15]. Targeting of microRNA (miRNA)-30a to the ubiquitin carrier protein 9 (Ubc9) was reported to mediate the stabilization of the PR domain-containing 16 (PRDM16) protein, which participated in the maintenance of classical BAs and the browning process of WAs [16]. The presence of truncated PGC-1α isoforms generated from the alternatively-spliced transcripts was characterized to enhance the mitochondrial respiration in active BAs [17]. In our previous studies, brown adipogenesis-induced expression of miR-485 was demonstrated to lessen the repressive influence of the serine/arginine-rich splicing factor protein kinase 1 (SRPK1) on BAs-associated splicing events [18]. Upregulated expression of the RNA-binding motif protein 4a (RBM4a) programmed multiple BAT-related AS events, which were relevant to the development or metabolic signatures of brown adipocytes [19].

In this study, we demonstrated that BBR treatment lessened the activity of the *c13orf25* promoter, which drove the transcription of *miR-92a*. Reduction of *miR-92a* resulted in an increase in RBM4a protein, which activated the brown adipogenesis-related gene expressions and splicing networks. Overexpressing *miR-92a* or RBM4a targeting interfered with the effect of BBR treatment on enhancing brown adipogenesis. Accordingly, our findings proposed a transcription-coupled post-transcriptional pathway that participated in the BBR-induced brown adipogenesis.

## 2. Materials and Methods

### 2.1. Cell Culture, In Vitro Differentiation, and Chemicals

Mouse 3T3-L1 pre-adipocytes were cultured in the growth medium (GM), composed of Dulbecco’s modified Eagle medium (DMEM; Invitrogen, Carlsbad, CA, USA), supplemented with 10% fetal bovine serum (FBS; Invitrogen). To induce in vitro browning, 3T3-L1 cells were shifted to the induction of DMEM medium, supplemented with 20% FBS, 0.5 mM IBMX (Invitrogen), 12.7 μM dexamethasone (Invitrogen), and 10 µg/mL insulin (Invitrogen) for 48 h. The differentiating DMEM medium (DM) supplemented with 10% FBS, 10 μg/mL insulin, and 2 µM Rosiglitazone was replaced with the induction medium and replenished every 48 h for 4 days. BBR, purchased from Sigma–Aldrich (St. Louis, MO, USA), was dissolved in DMSO. 3T3-L1 cells maintained in the growth medium were treated with 5 μM BBR for 48 h.

### 2.2. miRNA-seq Analyses

In brief, total RNAs were extracted using the ReliaPrep RNA Miniprep System (Promega, Madison, WI, USA), according to the manufacturer’s protocol. Qualified RNAs at 8 μg with a high integrity number (RIN > 8.0) were subjected to library construction using the NEB Next Multiplex Small RNA Library Prep Set for Illumina (NEB, Ipswich, MA, USA), according to the manufacturer’s instructions, and sequenced on an Illumina Hi-Seq 4000 platform. Preliminary reads were trimmed, filtered, and aligned to the mouse reference genome (GRCm37), and small RNA high-quality reads were extracted and analyzed using the CLC Genomic Workbench (CLC bio, Aarhus, Denmark).

### 2.3. Plasmid Construction

To construct the mouse *miR-92a*-expressing vector, the primary sequence was the polymerase chain reaction (PCR), amplified and cloned into *Hind*III/*EcoR*V sites of the p3XFLAG-CMV14 vector (Sigma). The genomic element containing the *c13orf25* promoter was PCR amplified and cloned into *Hind*III/*EcoR*V sites of the pGL3-Basic vector (Sigma). To construct the pRL-mRBM4a reporter, the coding element of mouse RBM4a, containing the putative targeting sites of *miR-92a*, was PCR-amplified and then inserted into *Xba* I/*Not* I sites of the pRL-TK vector (Promega). The derived mutant plasmids harboring substituted nucleotides were all constructed using the QuikChange site-directed mutagenesis system (Stratagene, La Jolla, CA, USA).

### 2.4. Poly(A) Tailing of Small RNA

Small RNAs were prepared using the Reliaprep miRNA Miniprep System and subjected to the poly(A)-tailing using A-Plus Poly(A) polymerase (NEB), as per the user’s instruction manual. Total RNA (20 µg) was preheated to 65 °C for 10 min and then incubated with A-Plus Poly(A) polymerase (2 U) in a 100 μL mixture at 37 °C for 60 min. The reaction was stopped by storing the mixture in a −70 °C freezer. Tailed RNAs were extracted and precipitated using a phenol chloroform isoamyl alcohol (PCA)/ethanol method.

### 2.5. Transient Transfection, Reverse-Transcription (RT)-PCR and Quantitative (q)PCR Analyses

3T3-L1 cells were grown to 60%~70% confluence and transfected with the indicated plasmid, using Lipofectamine 3000, according to the manufacturer’s protocol (Invitrogen). Total RNAs were extracted using a PureLink RNA mini kit at 24 h post-transfection (Invitrogen). Total RNA (at 1 µg) was reverse-transcribed using SuperScriptase III (Invitrogen) in a 10 μL reaction and subjected to a PCR analysis with specific primer sets (Appendix A). Levels of GAPDH transcripts were used as the loading reference. The quantitative (q)PCR assay was conducted with SYBR green fluorescent dye and gene-specific primer sets (Appendix A) using an ABI One Step™ PCR machine (Applied Biosystems, Foster City, CA, USA). The relative mRNA level was quantitated by the ΔΔ-Ct method and normalized to the level of GAPDH mRNA.

### 2.6. Immunoblotting Assay

The immunoblot analysis was performed using an enhanced chemiluminescence (ECL) system (Millipore, Billerica, MA, USA), and results were monitored using the LAS-4000 imaging system (Fujifilm, Tokyo, Japan). The primary antibodies, including polyclonal anti-RBM4 (Santa Cruz Biotechnology, Santa Cruz, CA, USA), polyclonal anti-GAPDH (MDBio, Taipei, Taiwan), polyclonal anti-SP1 (Abcam, Taipei, Taiwan), monoclonal anti-SRPK1 (BD Transduction, San Jose, CA, USA), and monoclonal anti-MYC (MDBio) were applied in this study.

### 2.7. Oil-Red-O (ORO) Staining

In vitro cultured cells were washed twice with PBS and fixed with 4% paraformaldehyde for 10 min. Cells were equilibrated with 60% isopropanol for 5 min at room temperature and stained with a 0.3% filtered Oil-red-O (ORO) dye (Sigma) for 30 min. Stained cells were washed with distilled water and visualized using a TE2000-U microscope (Nikon, Tokyo, Japan). The ORO dye was extracted from stained cells using absolute isopropanol at room temperature for 2 h. The extracted ORO dye was quantified at 550 nm with a NanoDrop 2000 spectrophotometer (Thermo Scientific, Waltham, MA, USA).

### 2.8. Mitochondrial Analysis

In vitro cultured 3T3-L1 cells were shifted to DMEM, containing 100 nM MitoTracker Red FM (Invitrogen) for 30 min at 37 °C. The cells were washed with a prewarmed culture medium and visualized using an Olympus IX81 microscope (Tokyo, Japan). The signal strength of captured pictures was analyzed using TotalLab Quant.

### 2.9. Mitochondrial Respiration Assay

The oxygen consumption rate (OCR) of cultured cells was measured using a Seahorse XF24 bioanalyzer (Seahorse Bioscience, North Billerica, MA, USA). In brief, 2 × 10^4^ in vitro-cultured cells were seeded in each well of Seahorse XF24 plates with 250 µL of DMEM overnight. Prior to the analysis, cells were washed with an unbuffered medium and immersed in 675 µL of an unbuffered medium without CO_2_ for 1 h. The OCR was assessed in 8 min cycles, as recommended by Seahorse Bioscience. Basal and maximal OCRs and the spare respiratory capacity were recorded following injection of complex-specific substrates, including FCCP (2 μM), rotenone (2 μM), and oligomycin (2.5 µg/mL).

### 2.10. Bioinformatic Prediction of miRNA-RBM4a Heteroduplex

The publicly available algorithm, RNA22 V2.0 (https://cm.jefferson.edu/rna22/), was applied for the miRNA-RBM4a pair prediction. The sequence of mature miRNAs identified using miRNA-seq analyses was individually aligned with the sequence of RBM4a 3’UTR or coding region in the RNA22 V2.0 algorithm. The parameter setting for RNA22 is: Only 1 “un-paired” base is allowed within the seed size of 7, the minimum number of paired-up bases within predicted heteroduplex is 12, the maximum value for the folding energy in any reported heteroduplex  =  −14  kcal/mol, and 2 wobble paired bases are allowed in the seed region. The folding energy of the putative miRNA-RBM4a pair was further predicted using the RNAhybrid algorithm (http://bibiserv.techfak.uni-bielefeld.de/rnahybrid). The miRNA target pairs with low folding energy (<−14 kcal/mol) and convincing *p* value (<0.005) were subjected to the further investigation.

### 2.11. In Vitro Luciferase Reporter Assay

3T3-L1 cells were seeded in six-well plates (2 × 10^5^ cells/well), 24 h prior to transfection. The transfection mixture contained 0.5 µg of the engineered Firefly or Renilla luciferase reporters, 1 µg of the effector expression vector, and 0.5 µg of the pRL-TK or pGL3-Basic vector (Promega) as the control. After 24 h, transfectants were lysed using the passive lysis buffer, and cell debris was removed after centrifugation. Activities of the firefly and Renilla luciferases were measured using a dual-luciferase assay kit (Promega) and the Synergy HT multi-mode microplate reader (BioTek, Winooski, VT, USA).

### 2.12. Statistical Analyses

Analysis of variance (ANOVA) and student *t*-tests were performed to determine the significance of previous experiments. *p* < 0.05 was considered statistically significant.

## 3. Results

### 3.1. BBR Induces Brown-Adipogenic Signatures of 3T3-L1 Cells

Development of beige adipogenesis was extensively studied by using an in vitro culture system, including mouse fibroblast 3T3-L1 cells [20,21,22]. Rosiglitazone treatment-induced beige adipogenesis was simultaneously conducted for evaluating the impact of BBR stimulus on the browning of 3T3-L1 cells in this study [23,24]. The accumulation of oil droplets was noted in 3T3-L1 cells cultured in the BBR-supplemented growth medium using Oil red-O (ORO) staining (Figure 1a, upper, GM+BBR) compared to that of mock-treated cells (Figure 1a, upper, GM). A similar phenomenon was reproduced in 3T3-L1 cells cultured in a Rosiglitazone-supplemented differentiating medium (Figure 1a, upper, DM). Quantitative results of the extracted ORO dye illustrated that more oil droplets were accumulated in BBR-treated cells than in the Rosiglitazone-treated differentiating cells (Figure 1a, bar chart). The presence of the differentiating medium or BBR treatment enhanced the mitochondriogenesis of 3T3-L1 cells (Figure 1a, lower, GM + BBR, and DM) compared to that of non-differentiating cells cultured in the growth medium (Figure 1a, lower, GM). The increases in beige adipocyte-specific factors (*CITED1*, *BMP7*, *PRDM16,* and *UCP1*) with concomitant decreases in WA-related factors (*Resistin*) were identified in BBR-treated 3T3-L1 cells (Figure 1b, lane 2) or differentiating cells (Figure 1b, lane 3) compared to those of vehicle-treated cells (Figure 1b, lane 1). Results of the qPCR assays showed similar expression profiles of these adipocyte-related factors in parallel experiments (Appendix A). Presence of BBR alone or the Rosiglitazone-supplemented differentiating medium led to increases in the basal oxygen consumption rate (OCR), maximal OCR, and ATP production of treated 3T3-L1 cells compared to those of vehicle-treated cells (Figure 1c). Collectively, these results consistently validated the impact of BBR on inducing the beige adipogenic signatures of 3T3-L1 cells.

### 3.2. BBR-Enhanced BA-Related Splicing Events in 3T3-L1 Cells

In addition to transcriptional regulation, alternative splicing (AS) constituted another mechanism to meet the requirement for the development of BAs [17,18,19]. RT-PCR results showed that splicing profiles of BA-related *PRDM16*, *LIPIN1*, and *PKM* genes were reprogrammed with the BBR treatment. As shown in Figure 2a, the relatively high levels of *PRDM16^+ex16^*, *LIPIN1*^−ex6^, and *PKM2* transcripts were identified in mock-treated cells (Figure 2a, lane 1). Treatment of BBR alone drove the shift of these splicing events to *PRDM16^−ex16^*, *LIPIN1*^+ex6^, and *PKM1* transcripts, encoding functional isoforms in mature BAs (Figure 2a, lane 2) [25,26], which was identified as well in differentiating 3T3-L1 cells (Figure 2a, lane 3). RBM4a and SRPK1 proteins were previously documented to exhibit the opposite influence on the splicing profiles of *PRDM16*, *LIPIN1*, and *PKM* genes during brown adipogenesis [18]. Results of RT-PCR, qPCR analyses (Figure 2b and bar chart), and the immunoblotting assay (Figure 2c) consistently demonstrated an increase in RBM4a with a concomitant decrease in SRPK1 transcripts and the encoding protein in BBR-treated or differentiating 3T3-L1 cells (Figure 2b,c, lanes 2 and 3) compared to those of non-differentiating cells (Figure 2b,c, lane 1). These results showed the influence of BBR treatment on modulating the BA-related splicing network in 3T3-L1 cells.

### 3.3. BBR Treatment Modulates the Expression Profiles of miR-92a and miR-485

An increase in *miR-485* was reported to downregulate SRPK1 expression, which functioned as the repressor toward brown adipogenesis [18]. We wondered whether upregulated RBM4a expression was relevant to miRNA expression profiles during beige adipogenesis. Results of the transcriptome analyses showed decreased expressions of *miR-92a-5p* in the BBR-treated (Figure 3a, GM+BBR) or differentiating 3T3-L1 cells (Figure 3a, DM) compared to that of mock-treated cells (Figure 3a, GM). RT-PCR and qPCR results validated a decrease in *miR-92a-5p* with a concomitant increase in *miR-485* levels in differentiating 3T3-L1 cells (Figure 3b, lane 2; Figure 3c, black bar) or BBR-treated cells (Figure 3b, lane 3; Figure 3c, gray bar) compared to those of non-differentiating cells (Figure 3b, lane 1; Figure 3c, white bar). The folding energy of two heteroduplexes composed of *miR-92a-5p* and the RBM4a transcripts were estimated using the RNA22 V2.0 (https://cm.jefferson.edu/rna22/), RNAhybrid (http://bibiserv.techfak.uni-bielefeld.de/rnahybrid), and miRmap (http://mirmap.ezlab.org/app/) algorithms (Figure 3d). The inverse relevant of *miR-92a* expressions to the metabolic activity of human BATs was documented in a previous study [27]. These results suggested the potential influence of reduced *miR-92a-5p* on RBM4a expressions during beige adipogenesis.

### 3.4. BBR-Modulated Transcriptional Regulation Reduces miR-92a Expression

The transcription of the *miR-17-92a* cluster comprising six miRNAs was driven by the *C13orf25* promoter, of which the activity was modulated with the SP1 and c-MYC expression profiles in malignancies [28]. Results of the in vitro reporter assays showed that the presence of the differentiating medium (Figure 4a, gray bar, DM) or BBR treatment (Figure 4a, gray bar, GM+BBR) led to a decrease in *c13orf25* promoter activity compared to the mock-treated cells (Figure 4a, gray bar, GM). A decrease in the SP1 transcript and encoding protein with a concomitant increase in the MYC mRNA and encoding product was identified with BBR treatment (Figure 4b, lanes 3 and 6) or in the presence of the differentiating medium (Figure 4b, lanes 2 and 5) compared to the mock-treated 3T3-L1 cells (Figure 4b, lanes 1 and 4). To further demonstrate the impact of SP1 or MYC on *c13orf25* promoter activity, the derived mutant containing substituted nucleotides within the SP1-binding site (Mut1) or MYC-binding site (Mut2) were constructed (Figure 4c, upper) and subjected to an in vitro reporter assay. The results showed that disruption of the SP1-binding site (Mut1) reduced the activity of the *c13orf25* promoter (Figure 4c, GM, gray bar), whereas nucleotide substitutions within the MYC-binding site (Mut2) reversely strengthened the *c13orf25* promoter activity (Figure 4c, GM, black bar) compared to that of the wild-type (WT) *c13orf25* promoter in mock-treated cells (Figure 4c, GM, white bar). The presence of the differentiating medium or BBR exhibited a similar effect on further reducing the activity of WT or Mut1 promoters (Figure 4c, DM and GM+BBR, white and gray bar) compared to those of non-differentiating cells (Figure 4c, GM, white and gray bar). In contrast, disruption of the MYC responsive site lessened the repressive effect of the differentiating medium or BBR treatment on the activity of the Mut2 reporter (Figure 4c, DM and GM+BBR, black bar) compared to that of non-differentiating cells (Figure 4c, GM, black bar). These results suggested that BBR-modulated expressions of SP1 and MYC protein constituted a synchronous mechanism for regulating the *c13orf25* promoter activity.

### 3.5. Overexpression of miR-92a-5p Leads to a Decrease in RBM4a Expression

To validate the targeting specificity of mice *miR-92a-5p* to RBM4a transcripts, the WT miR-92a-1 expression vector was constructed and mutated by guanine-to-thymidine or adenine-to-cytosine substitutions at the second nucleotide of the seed sequence of *miR-92a-2-5p* and *miR-92a-2-3p*. The Renilla luciferase reporter harboring two *miR-92a* targeting sites within the mice RBM4a open reading frame was established, and the derived mutant was constructed by the thymidine-to-guanosine substitution in each (Mut 1 or Mut 2) or both targeting sites (Mut 3), as shown in Figure 5b. The impact of heteroduplexes composed of *miR-92a-5p* and RBM4a transcripts on RBM4a expression was validated using in vitro reporter and immunoblotting assays. As shown in Figure 5a, a decrease in the RBM4a transcript and coding protein was identified with the overexpression of *miR-92a-5p* or the *miR-92a-m3p* (Figure 5a, lanes 2, 4, 6, and 8) compared to that of the scramble miRNA- or *miR-92a-m5p* overexpressing cells (Figure 5a, lanes 1, 3, 5, and 7). Results of the in vitro reporter assays showed a reduced activity of the pRL-mRBM4a reporter compared to that of the pRL-TK vector in mock-treated 3T3-L1 cells (Figure 5b, white bar). Disruption of each *miR-92a-5p* targeting site restored the activity of the mutant reporter (Fig. 5B, white bar, Mut1, and Mut2). The absence of the *miR-92a-5p* targeting site within the inserted RBM4a ORF exhibited no effect on the activity of the Mut3 reporter as compared to the pRL-TK vector in non-differentiating cells (Figure 5b, white bar, Mut3). Nevertheless, the activities of the pRL-mRBM4a, Mut1, and Mut2 reporters were upregulated in the differentiating cells or BBR-treated cells (Figure 5b, gray bar and black bar) compared to those of the non-differentiating cells (Figure 5b, white bar). Overexpression of *miR-92a-5p* or *miR-92a-m3p*, but not *miR-92a-m5p*, led to the decreases in pRL-mRBM4a activities compared to that of the scramble miRNA-transfected cells (Figure 5c, gray bar). Disruption of *miR-92a-5p* targeting sites relieved the repressive effect of overexpressing *miR-92-5p* on the activity of mutant reporters (Figure 5d, gray bar, Mut1 to Mut3) compared to that of WT reporter (Figure 5d, gray bar, pRL-mRBM4a). These results illustrated the impact of *miR-92a-5p* on regulating the expression of RBM4a.

### 3.6. Overexpression of miR-92a-5p Interferes with the Influence of BBR on Beige Adipogenesis

To evaluate the influence of *miR-92a* on BBR-induced in vitro browning, 3T3-L1 cells were respectively transfected with the expressing vector generating scramble miRNA, WT *miR-92a-5p*, or *miR-92a-m5p* and subjected to BBR treatment. Results of the ORO staining showed that less lipid droplet-containing cells were observed with the overexpression of WT *miR-92a-5p* (Figure 6a, *miR-92a-5p*) compared to that of *miR-92a-m5p* overexpressing cells (Figure 6a, *miR-92a-m5p*) or scramble miRNA-overexpressing cells (Figure 6a, Scramble miR) with BBR treatment. Overexpression of *miR-92a-m5p* exhibited no effect on the expression profiles of BAs- or beige cell-related transcripts in the transfected cells cultured in the growth medium, BBR-supplemented medium, or differentiating medium compared to those of scramble miRNA-transfected cells (Figure 6b, Scramble and m5p). In contrast, overexpressing WT *miR-92a-5p* diminished the effect of the BBR or differentiating medium by enhancing the expressions of BA-related transcripts (Figure 6b, WT, GM + BBR and DM). The presence of overexpressing WT-*miR-92a-5p* interfered with the impact of BBR or the differentiating medium on the splicing profiles of *PRDM16*, *LIPIN1* and *PKM* (Figure 6c, lanes 4–6). In contrast, the presence of *miR-92a-m5p* or scramble miRNA had no effect on the splicing profiles of *PRDM16*, *LIPIN1*, or PKM transcripts that were reprogrammed with BBR treatment (Figure 6c, lanes 2 and 8) or the differentiating medium (Figure 6c, lanes 3 and 9) compared to those of the non-differentiating cells (Figure 6c, lanes 1 and 7). The WT *miR-92a*-*5p* or *miR-92a-m3p*-overexpressing cells exhibited relatively low levels of maximal OCR and ATP production (Figure 6d, GM, orange and purple bar) compared to those of the scramble miRNA-transfectants or *miR-92a-m5p*-overexpressing cells cultured in the growth medium (Figure 6d, GM, blue and green bar). The influence of BBR treatment or the differentiating medium on enhancing maximal oxygen consumption or ATP production was abolished in the presence of overexpressing WT *miR-92a-5p* or *miR-92a-m3p* (Figure 6d, GM + BBR and DM, orange and purple bar), but not *miR-92a-m5p* (Figure 6d, GM + BBR and DM, green bar). Taken together, upregulated *miR-92a* expression diminished the BBR-enhanced thermogenic signatures in 3T3-L1 cells.

### 3.7. Depletion of RBM4a Diminishes the Effect of BBR on Brown Adipogenesis

To validate the correlation between RBM4a and BBR-induced in vitro browning, the endogenous RBM4a was depleted by introducing the targeting vector in 3T3-L1 cells (Figure 7a, lanes 2 and 3). Results of ORO staining showed that depletion of endogenous RBM4a diminished the effect of BBR on inducing lipid accumulation (Figure 7a, right, shRBM4a-1) compared to that of empty vector-transfected cells (Figure 7a, right, pSilencer). Depletion of RBM4a lessened the effect of BBR treatment or the differentiating condition on enhancing the expressions of BA-related transcripts (Figure 7b, GM+BBR and DM, shRBM4a) compared to those of empty vector-transfected cells (Figure 7b, GM+BBR and DM, vector). Knockdown of RBM4a reduced the impact of BBR treatment or the differentiating medium on shifting BA-related splicing events to *PRDM16^−ex16^*, *LIPIN1^+ex6^*, and *PKM1* transcripts (Figure 7c, lanes 4–9) compared to those of empty vector-transfectants cultured in the same conditions (Figure 7c, lanes 1–3). RBM4a-depleted cells exhibited reduced activities in maximal OCRs and ATP production (Figure 7d, orange bar) compared to those of empty vector-transfected cells (Figure 7d, blue bar) cultured in the growth medium, or in the differentiating medium, or with BBR treatment. These results suggested that the abundance of RBM4a was relevant to the impact of BBR on the browning of 3T3-L1 cells.

## 4. Discussion

Inducible BAs are considered a therapeutic strategy for combating obesity due to the advancing lipolysis and energy expenditure [29,30]. The benefits of BBR to promote metabolism and lipolysis of adipocytes evoked wide interest in food compounds as a treatment strategy for obesity. Our results demonstrated the emerging transcription-coupled post-transcriptional mechanism involved in BBR-mediated beige adipogenesis.

Potential applications of BAs to combat obesity have sparked great interest in inducing trans-differentiation of mature WAs [29]. Although well-differentiated WAs were demonstrated to undergo direct conversion into BA-like cells [31], the origin of beige cells is debatable [32]. The presence of a phytochemical, including BBR, committed additional progenitor or in vitro cultured cells toward the development of BA-like cells [33,34]. BBR administration mediated an increase in the mass of beige adipocytes in subcutaneous and inguinal WATs in small rodents [9]. In these studies, BBR treatment mediated increases in BA-specific genes, such as *PGC-1α* and *PPARγ*, which enhanced brown adipogenic signatures through diverse pathways [35]. BBR treatment also activated a brain and muscle Arnt-like 1 (Bmal1)-mediated pathway, which led to an increase in fibroblast growth factor 21 (FGF21), functioning as another key regulator of brown adipogenesis [36,37]. Although the impact of BBR on post-transcriptional control, particularly alternative splicing, throughout brown adipogenesis was largely unknown, BBR-manipulated alternative splicing events were relevant to the carcinogenic signatures, such as active proliferation and anti-apoptosis. [38,39]. In this study, BBR treatment altered protein levels of RBM4a and SRPK1 by reprogramming miRNA expression profiles, which drove the shift of *PRDM16*, *LIPIN1*, and *PKM* transcripts, encoding BA-specific isoforms involved in transcriptional regulation, lipid storage, and metabolic activity [18,19,25]. These results demonstrated that BBR enhanced beige adipogenesis through a complex pathway, composed of transcriptional and post-transcriptional regulation.

In addition to alternative splicing, miRNA constituted another regulatory mechanism that participates in the development of BAs. For instance, the abundances of *miR-26* family members were relevant to the brown adipogenic signatures of human adipocytes [40]. An inverse correlation of exosome-containing *miR-92a* with thermogenic activity was noted in an animal model, which is linked to BAs and beige adipocytes [27]. In contrast, overexpression of *miR-92a* was reported to facilitate the development of WAs [41]. These results implied the discriminative impacts of *miR-92a* on the development of WAs and BAs. An increase in Argonaute-2 (AGO2), the critical miRNA mediator, throughout development of BATs was recently identified, which was highly relevant to the thermogenic activity of mature BAs [42]. These findings further suggested the importance of the miRNA-regulated pathway for the development of BATs. In this study, the presence of BBR was shown to modulate the expressions of SP1 and MYC, which determined the activity of the *c13orf25* on the driving transcription of the *miR-17/92a* cluster. A decrease in the *miR-92a* lessened its repressive effect on RBM4a expression, which in turn facilitated the brown adipogenesis (Figure 7e). These results disclosed the emerging influence of BBR on post-transcriptional control throughout the browning of WAs.

## Figures and Tables

**Figure 1 cells-08-00632-f001:**
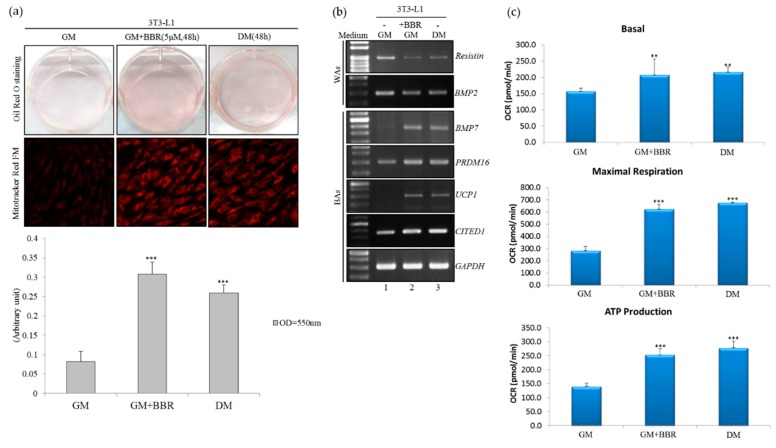
Berberine (BBR) treatment-enhanced beige adipogenesis of 3T3-L1 cells. (**a**) 3T3-L1 cells were cultured in the growth medium, the BBR-supplemented growth medium (5 µM), or the Rosiglitazone-supplemented differentiation medium for 48 h, followed by Oil-Red-O staining (upper) and Mitotracker Red FM staining (lower). The bar graph presents the spectrophotometric analysis of Oil-Red-O dye extracted from the stained cells. (**b**) Total RNAs extracted from in vitro cultured cells were subjected to RT-polymerase chain reaction (PCR) assays using specific primer sets, as listed in Appendix A. (**c**) 3T3-L1 cells were cultured in the growth medium, the BBR-supplemented growth medium (5 µM), or the Rosiglitazone-supplemented differentiation medium for 48 h, followed by the bioenergetic analyses using a Seahorse XF24 Bioanalyzer (*n* = 4). The bar graphs present the basal oxygen consumption rate (OCR), maximal OCR, and ATP production rate of cultured cells. Quantitative results are presented as the mean ± SD (*n* = 4), and statistical significance was calculated using the student unpaired *t*-test (** *p* < 0.01; *** *p* < 0.005).

**Figure 2 cells-08-00632-f002:**
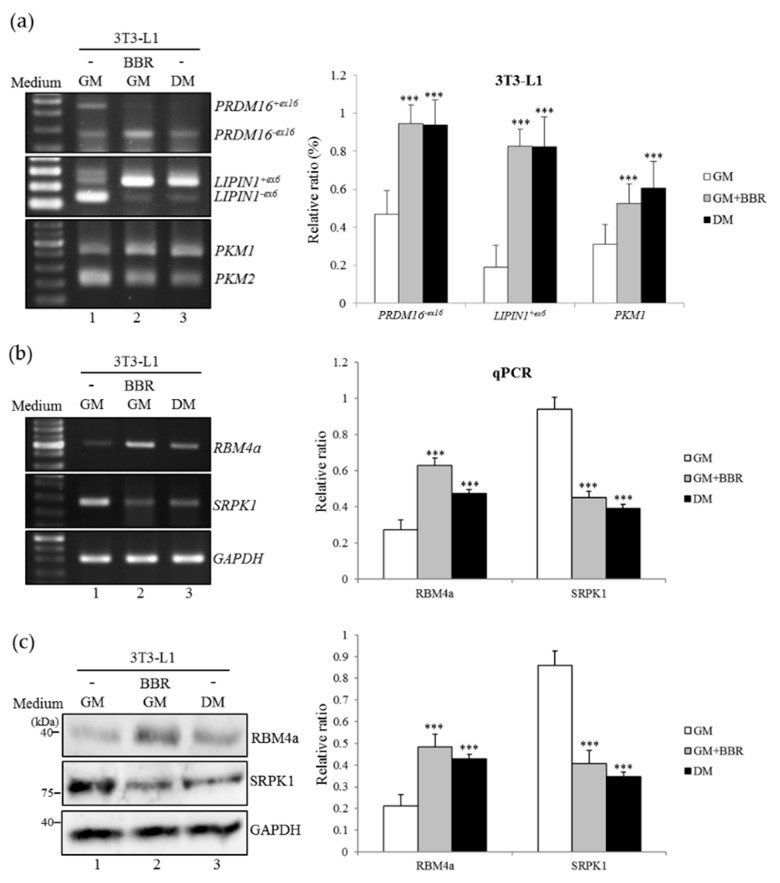
Berberine treatment reprograms the brown adipocyte-related splicing network. (**a**) 3T3-L1 cells were cultured in the growth medium, BBR-supplemented growth medium (5 µM), or Rosiglitazone-supplemented differentiation medium for 48 h. Total RNAs were extracted and subjected to RT-PCR analyses with indicated primer sets listed in Appendix A. The bar graph shows the percentage isoform index (PII), which presents relative levels of spliced transcripts. (**b**,**c**) Total RNAs and cell lysates prepared from the parallel experiments were subjected to RT-PCR, qPCR, and immunoblot assays using specific primer sets, listed in Appendix A, and specific antibodies. Quantitative results are presented as the mean ± SD (*n* = 4), and statistical significance was calculated using the student unpaired *t*-test (*** *p* < 0.005).

**Figure 3 cells-08-00632-f003:**
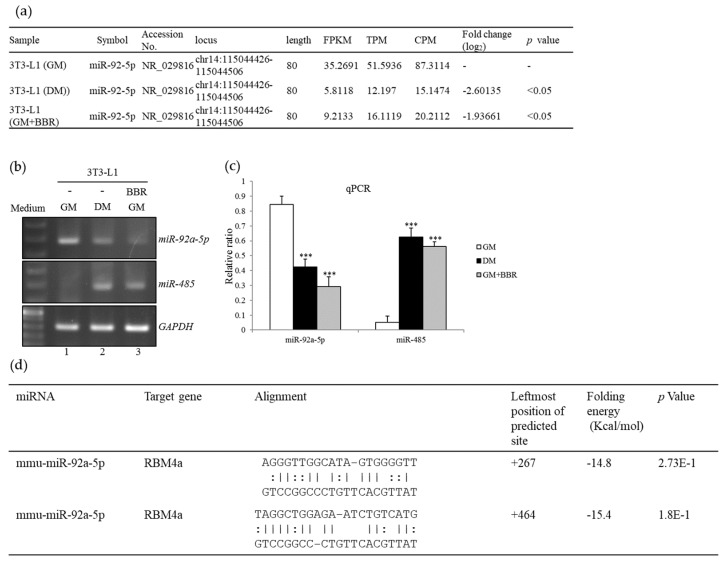
Berberine treatment reprogram expression profiles of miRNAs. (**a**) Results of miRNA-seq assays were analyzed using the CLC genomics workbench (CLC, GWB). Analytical results showed discriminative expressions of *miR-92a-5p* in 3T3-L1 cells cultured in the growth medium (GM), Rosiglitazone-supplemented DM), or BBR-supplemented growth medium (GM+BBR). (**b**,**c**) Total RNAs prepared from the parallel experiments were subjected to RT-PCR and qPCR assays with specific primer sets as listed in Appendix A. (**d**) The folding energy of the *miR-92a*-RBM4a heteroduplex was assessed by aligning the corresponding sequences in the RNA22 2.0 algorithm. Quantitative results are presented as the mean ± SD (*n* = 4), and statistical significance was calculated using the student’s unpaired *t*-test (*** *p* < 0.005).

**Figure 4 cells-08-00632-f004:**
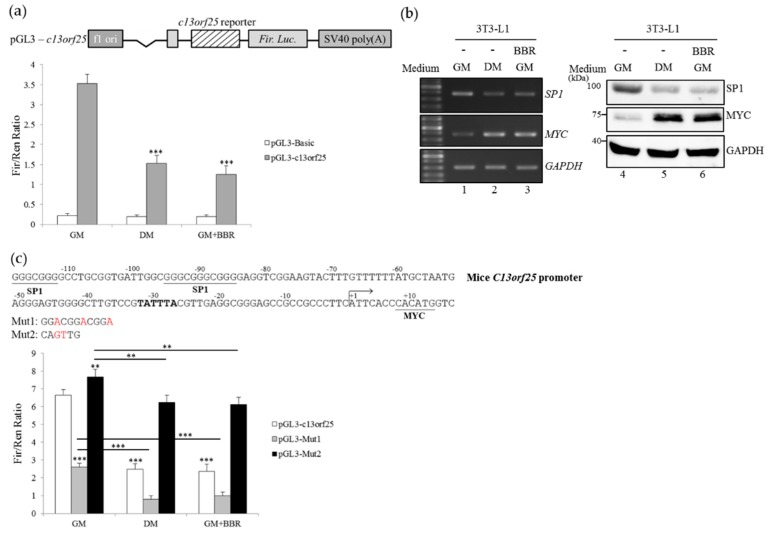
Berberine treatment reduces the activity of the *c13orf25* promoter. (**a**) The intact pGL3-*c13orf25* reporter (upper) or pGL3-basic vector were respectively cotransfected with the pRL-TK vector into 3T3-L1 cells which were cultured in the growth medium (GM), Rosiglitazone-supplemented DM, or BBR-supplemented growth medium (GM+BBR). In vitro reporter assays were conducted after 48 h. (**b**) Total RNA and cell lysates were prepared from 3T3-L1 cells cultured in the growth medium (GM), differentiating medium (DM), or BBR-supplemented growth medium (5 µM). Expression profiles of SP1 and MYC transcripts and proteins were analyzed using RT-PCR and immunoblot assays with specific primer sets and antibodies. (**c**) The diagram presents the sequence of the proximal region of the *c13orf25* promoter (upper). The pGL3-*c13orf25* reporter and the derived mutants was respectively co-transfected with the pRL-TK control vector into 3T3-L1 cells cultured in the growth medium (GM), the Rosiglitazone-supplemented differentiating medium (DM), or the growth medium supplemented with BBR (GM+BBR). In vitro reporter assays were conducted after 48 h. The bar graph presents the relative *Firefly* luciferase activity normalized to *Renilla* luciferase activity. Quantitative results are presented as the mean ± SD (*n* = 4), and the statistical significance was calculated using the student’s unpaired *t*-test (** *p* < 0.01; *** *p* < 0.005).

**Figure 5 cells-08-00632-f005:**
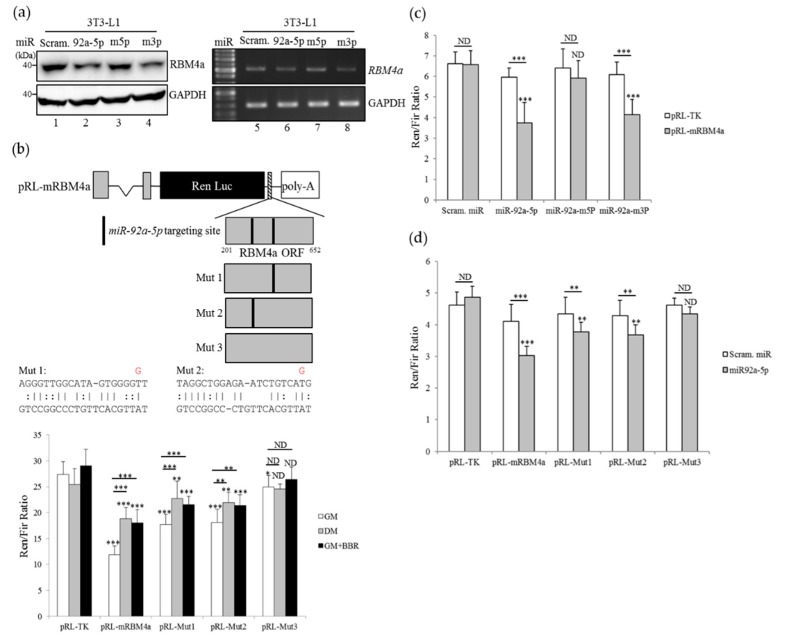
*miR-92a* reduces the expression of RBM4a by targeting the RBM4a coding region. (**a**) Total RNAs and cell lysates were extracted from 3T3-L1 cells transfected with a scramble miRNA, or an expressing vector generating wild-type (WT) *miR-92a-5p* or mutant *miR-92a*. RT-PCR and immunoblot assays were conducted using specific primer sets and antibodies. (**b**) The scheme presents the *Renilla* luciferase reporter harboring two *miR-92a-5p*-targeting sites within the RBM4a coding region and the derived mutants. The pRL-mRBM4a reporter or the derived mutant was respectively co-transfected with the pGL3-basic vector into 3T3-L1 cells cultured in the growth medium (GM), the Rosiglitazone-supplemented differentiating medium (DM), or the BBR-supplemented growth medium (GM+BBR). In vitro reporter assays were conducted after 48 h. (**c**) The pRL-*mRBM4* reporter or pRL-TK vector was respectively co-transfected with the expressing vector of the scramble miRNA, *miR-92a-5p*, or the mutant *miR-92a* into 3T3-L1 cells. (**d**) The scramble miRNA or *miR-92a*-*5p* expressing vector was respectively co-transfected with the pRL-*mRBM4* or the derived reporter into 3T3-L1 cells. In vitro reporter assays were conducted post-transfection 24 h. The bar graph presents the relative reporter activity normalized to control luciferase activity. Statistical significance was determined using the student’s unpaired *t*-test (* *p* < 0.05; ** *p* < 0.01; *** *p* < 0.005).

**Figure 6 cells-08-00632-f006:**
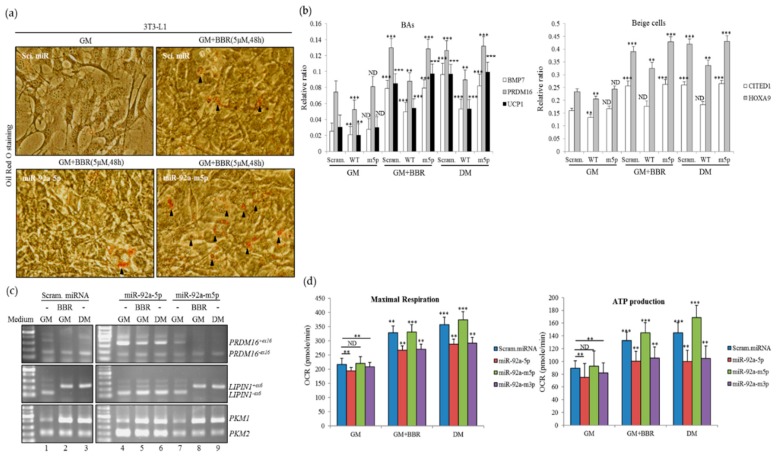
Overexpression of *miR-92a* interferes with BBR-mediated beige adipogenesis. (**a**) 3T3-L1 cells were respectively transfected with the expressing vector of the scramble miRNA, or the *miR-92a-5p*, or the mutant *miR-92a*, followed by culturing in the growth medium (GM) or the BBR-supplemented growth medium (5 µM) for 48 h. Lipid droplets (arrowhead) were monitored using Oil-Red-O staining. (**b**,**c**) Total RNAs were prepared from the transfected cells cultured in the growth medium (GM), or the Rosiglitazone-supplemented differentiating medium (DM), or BBR-supplemented growth medium. QPCR and RT-PCR assays were performed using specific primer sets, as listed in Appendix A. (**d**) 3T3-L1 cells were subjected to the parallel experiments as described in last panel, followed by bioenergetic analyses using a Seahorse XF24 Bioanalyzer (*n* = 4). The bar graph presents quantitative results as the mean ± SD. Statistical significance was calculated using the student’s unpaired *t*-test (** *p* < 0.01; *** *p* < 0.005).

**Figure 7 cells-08-00632-f007:**
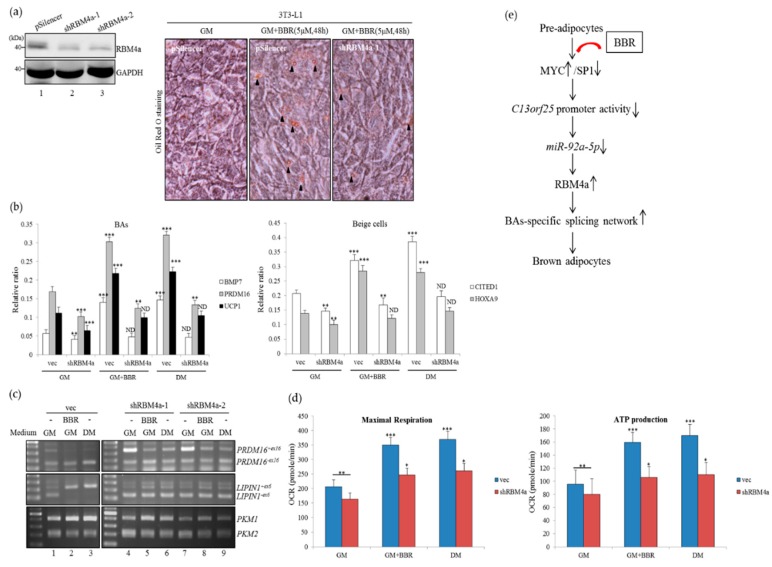
Depletion of RBM4a diminished the influence of BBR on beige adipogenesis. (**a**) 3T3-L1 cells were respectively transfected with the empty vector or targeting vectors against mRBM4a (shRBM4a-1 and 2). Total cell lysates were extracted 24 h post-transfection and subjected to immunoblot assays with indicated antibodies. Oil-Red-O staining was performed to monitor the lipid accumulation (arrowhead) in 3T3-L1 cells applied to the parallel transfection, followed by culturing in the growth medium (GM) or supplementation with BBR (GM+BBR). (**b**,**c**) Total RNA samples were prepared from 3T3-L1 cells applied to the parallel transfection, followed by culturing in the growth medium (GM), or the Rosiglitazone-supplemented differentiating medium (DM), or the BBR-supplemented growth medium (GM+BBR). QPCR and RT-PCR assays were performed using specific primer sets, as listed in Appendix A. (**d**) 3T3-L1 cells were applied to the parallel experiments and subjected to bioenergetic analyses using a Seahorse XF24 Bioanalyzer (*n* = 4). The bar graph presents quantitative results as the mean ± SD. Statistical significance was calculated using the student’s unpaired *t*-test (* *p* < 0.05; ** *p* < 0.01; *** *p* < 0.005). (**e**) In this study, BBR treatment was demonstrated to reduce the activity of the *c13orf25* promoter by modulating SP1 and MYC expressions, which led to a decrease in the *miR-92a* level. Consequently, BBR treatment relieved the repressive effect of *miR-92a-5p* on RBM4a expression, which facilitated brown adipogenesis through post-transcriptional regulation.

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
