# Peer review of "Berberine Promotes Beige Adipogenic Signatures of 3T3-L1 Cells by Regulating Post-transcriptional Events"

_cells, 2019, doi:10.3390/cells8060632_

Round 1

Reviewer 1 Report

The present study entitled to “Berberine promotes beige adipogenic signatures of 3T3-L1 cells by regulating post-transcriptional events” investigated the effect of berberine on beige/brwon adipogenesis through post-transcriptional control. It is very interesting and informative, but there are several points that should be addressed

1.  Detail and clear description seem to be necessary in the section of 2.1. cell culture, in vitro differentiation and chemicals. What is GM, GM+BBM, and DM? When and how to induce differentiation and to treat berberine needs to be clearly demonstrated. 

2.  To demonstrate berberine-induced brown fat like phenotype, more specific methods should be used. There are some examples as follows; immunofluorescence staining with UCP-1 antibody, Mitotacker red staining, transmission electronic microscopy, and gene/protein expression related to mitochondria biogenesis and function. 

3. Please explain what miR-92a-5p, miR-92a-m5p and miR-92a-m3p and what pRL-mRBM4a, Mut1, and Mut2 are before results in the section of 3.5. Overexpression of miR-92a-5p leads to a decrease in RBM4a expression.

4.  What is the relationship between miR-92-5p and miR-485? Do you think whether BBR-increased miR-485 also influences SRPK1-related splicing network

5. Reference 10 was not relevant to berberine. I suggest used references need to be rechecked out.

Author Response

Response to Reviewers' Comments

(Manuscript Number: Cells-521082)

Dear Editor:

We thank you for your response and for allowing revision of our manuscript (Cells-521082; Berberine promotes beige adipogenic signatures of 3T3-L1 cells by

regulating post-transcriptional events). The manuscript was revised in line with the valuable suggestions and comments of all reviewers. The revised manuscript was edited by a native English-speaking professional editor prior to submission. We hope that the revised manuscript achieves reviewer satisfaction. Our point-by-point responses to all specific reviewer comments, suggestions, and queries are as follows.

REVIEWER(S)' COMMENTS

Reviewer #1: The present study entitled to “Berberine promotes beige adipogenic signatures of 3T3-L1 cells by regulating post-transcriptional events” investigated the effect of berberine on beige/brwon adipogenesis through post-transcriptional control. It is very interesting and informative, but there are several points that should be addressed

Response to Reviewer Comments

1.  Detail and clear description seem to be necessary in the section of 2.1. cell culture, in vitro differentiation and chemicals. What is GM, GM+BBM, and DM? When and how to induce differentiation and to treat berberine needs to be clearly demonstrated. 

Response:

The experimental protocol regarding in vitro differentiation and berberine treatment of 3T3-L1 cells was further described according to the comment of reviewer. (please see Materials and methods sec. 2.1, line 64-73)

2.  To demonstrate berberine-induced brown fat like phenotype, more specific methods should be used. There are some examples as follows; immunofluorescence staining with UCP-1 antibody, Mitotacker red staining, transmission electronic microscopy, and gene/protein expression related to mitochondria biogenesis and function. 

Response:

Mitotracker Red FM staining was conducted to demonstrate berberine-induced brown fat like phenotype according to the comment of reviewer (please see Figure 1a and line 130-134; 180-182).

3. Please explain what miR-92a-5p, miR-92a-m5p and miR-92a-m3p and what pRL-mRBM4a, Mut1, and Mut2 are before results in the section of 3.5. Overexpression of miR-92a-5p leads to a decrease in RBM4a expression.

Response:

The manuscript was revised according to the comment of reviewer (please see Results sec. 3.5, line 298-306).

4. What is the relationship between miR-92-5p and miR-485? Do you think whether BBR-increased miR-485 also influences SRPK1-related splicing network.

Response:

MiR-92a and miR-485 was proposed to constitute coordinate pathway participated in the development of neuron (Muñoz-Llanos M et al, Front. Mol. Neurosci. 2018, 11, 251.), whereas the correlation between miR-92a and miR-485 involved in adipogenesis is not documented. Nevertheless, BBR-mediated increase in miR-485 exhibits potential impact on SRPK1-regulated splicing network.

5. Reference 10 was not relevant to berberine. I suggest used references need to be rechecked out.

Response:

The appropriate reference is cited in the revised manuscript (please see Ref. 10).

Reviewer 2 Report

The manuscript describes an investigation of the effects of berberine on adipocytes browning, considering the post-trascriptional modifications. The subject of the manuscript could fall within the scope of the Journal and interesting to publish, however it would be help if the author could provide more information, as the following.

The manuscript describes an investigation of the effects of berberine on adipocytes browning, considering the post-trascriptional modifications. The subject of the manuscript could fall within the scope of the Journal and interesting to publish, however results are.  The text should be improved by having input from a native English speaker. Authors should introduce more extensively post-trascriptional regulation in order to facilitate the reader the following results and aim of work. Material and Methods are lacking of many information and should be re-written.

Author Response

Response to Reviewers' Comments

(Manuscript Number: Cells-521082)

Dear Editor:

We thank you for your response and for allowing revision of our manuscript (Cells-521082; Berberine promotes beige adipogenic signatures of 3T3-L1 cells by

regulating post-transcriptional events). The manuscript was revised in line with the valuable suggestions and comments of all reviewers. The revised manuscript was edited by a native English-speaking professional editor prior to submission. We hope that the revised manuscript achieves reviewer satisfaction. Our point-by-point responses to all specific reviewer comments, suggestions, and queries are as follows.

REVIEWER(S)' COMMENTS

Reviewer #2:

The manuscript describes an investigation of the effects of berberine on adipocytes browning, considering the post-trascriptional modifications. The subject of the manuscript could fall within the scope of the Journal and interesting to publish, however it would be help if the author could provide more information, as the following.

The English usage is often not correct and this makes difficult to understand the experiments and conclusion. The text should be improved by having input from a native English speaker. Authors should introduce more extensively post-trascriptional regulation in order to facilitate the reader the following results and aim of work. Material and Methods are lacking of many information and should be re-written.

Response to Reviewer Comments

The English usage is often not correct and this makes difficult to understand the experiments and conclusion. The text should be improved by having input from a native English speaker.

Response:

The manuscript was edited by native English-speaking professional editor prior to submission. The certification is provided as follows.

Abstract

Line 18: ‘reported to induce the beige adipogenesis of white adipocytes’ to induce conversion from white to beige….

whole transcriptome or miRNA seq?

Response:

The sentence was revised according to the comment of reviewer (please see line 17 and 18).

Introduction

Too short it does not properly introduce the following analysis.

Line 31: results

Line 32: executes

Line 35: why excess exercise? …

which were proposed…

Response:

1.       The introduction was revised according to the comment of reviewer (please see Introduction sec).

2.       The sentence was revised according to the comment or reviewer (please see line 31, 34 and 35).

Material and method

Line 63: the Rosiglitazone concentration is lacking and it is not clear the sentece ‘was replaced with differentiation medium (DM)’. Is it the positive control?

Line 88: PolyA polymerase concentration

GM is considered negative control? This have to be stated properly.

Which is the BBB concentration? How did you choose the concentration? Have you perform a MTT assay?

Response:

1.      The manuscript was revised according to the comment of reviewer (please see Materials and methods sec. 2.1, line 67-71). In vitro browning assays acted the positive control to evaluate the effect of BBR treatment on enhancing brown adipogenic signatures, whereas non-differentiating cells cultured in ordinary growth medium (GM) was considered the negative control in this study (please see line 65).

2.      The working concentration of BBR and polyA polymerase was provided in the revised manuscript (please line 72 and 98). The working concentration of BBR (5mM) for in vitro treatment was widely applied and had no influence on cell mortality (Jiang et al, Lipids Health Dis. 2016, 15, 214.; Zhang et al, PLoS One, 2015, 10, e0125667.).

Results

Lines 220 The results reported data obtained with method that were not mentioned in M&M

In figures why some statistical significances are in square or round bracket?

In figure 3d you mentioned RNA22 2.0 algoritm that in M&M is not described.

In the legend of Fig.4 you mentioned luciferase that in M&M is not reported.

Response:

1.      The analysis of miRNA-seq was described in the revised manuscript (please see line 81 and 82).

2.      The statistic significances were properly marked throughout the manuscript to clearly illustrate the results.

3.      The experimental protocol for prediction of miRNA-RBM4a heteroduplex was described in the revised manuscript (please see line 145-155).

4.      The experimental protocol for in vitro luciferase reporter assay was described in the revised manuscript (please see line 157-164).

Reviewer 3 Report

In this manuscript, Lin et al. identified a novel mechanism of BBR treatment-mediated beige adipogenesis of 3T3-L1 cells through transcription-coupled post-transcriptional regulation. Whilst this study is generally well performed, it is rather confusing in some places and the demonstrations are still poorly defined.

Points,

1. In page 2, line 63 (Materials and Methods section), there might be a typo error: ‘2  M Rosiglitazone’.

2. In the reference articles the authors presented as source of browning differentiation protocols, the differentiation period is more than 7 days. However, the authors incubated 3T3-L1 cells with differentiation medium for only 48 h. Generally, differentiation requires longer incubation than 48 h. It should be explained why they incubated the cells only 48 h.

3. In figure 1a, authors used oil red o staining to evaluate the impact of BBR on browning of 3T3-L1. However, the oil drop can be stained in white adipocytes as well as brown or beige adipocytes. The expression of white adipocytes marker should be assessed in this condition.

4. In several quantitative results with statistical significance, the mark of significance ‘*’ are put in parentheses. The meaning of parentheses should be demonstrated.

5. In materials and methods section, the authors stated that the relative mRNA level subjected to qPCR was normalized to that of GAPDH. But there are also GAPDH level in qPCR data (figure 2b and c, figure 3c). Is it right that qPCR was performed instead of quantifying RT-PCR data?

6. In figure 5, authors used miR-92a-m3p and miR-92a-m5p as mutant miR-92a. It is necessary to explain what they are. Additionally, miR-92a-m3p did not have resistance as a mutant unlike miR-92a-m5p. The meaning of these results should be addressed.

Author Response

Response to Reviewers' Comments

(Manuscript Number: Cells-521082)

Dear Editor:

We thank you for your response and for allowing revision of our manuscript (Cells-521082; Berberine promotes beige adipogenic signatures of 3T3-L1 cells by

regulating post-transcriptional events). The manuscript was revised in line with the valuable suggestions and comments of all reviewers. The revised manuscript was edited by a native English-speaking professional editor prior to submission. We hope that the revised manuscript achieves reviewer satisfaction. Our point-by-point responses to all specific reviewer comments, suggestions, and queries are as follows.

REVIEWER(S)' COMMENTS

Reviewer #3: In this manuscript, Lin et al. identified a novel mechanism of BBR treatment-mediated beige adipogenesis of 3T3-L1 cells through transcription-coupled post-transcriptional regulation. Whilst this study is generally well performed, it is rather confusing in some places and the demonstrations are still poorly defined.

Response to Reviewer Comments

1. In page 2, line 63 (Materials and Methods section), there might be a typo error: ‘2  M Rosiglitazone’.

Response:

I apologize for the mistake and this typo was corrected in the revised manuscript. (please see Materials and methods sec. 2.1, line 70)

2.  In the reference articles the authors presented as source of browning differentiation protocols, the differentiation period is more than 7 days. However, the authors incubated 3T3-L1 cells with differentiation medium for only 48 h. Generally, differentiation requires longer incubation than 48 h. It should be explained why they incubated the cells only 48 h.

Response:

In vitro browning of 3T3-L1 cells is conducted for 6 days in this study according to the experimental protocol in the revised manuscript (please see Materials and methods sec. 2.1, line 67-71). Nevertheless, the brown adipogenic signatures of differentiating 3T3-L1 cells were characterized in the presence of differentiating medium (DM) for 48h using multiple functional assays.

3.  In figure 1a, authors used oil red o staining to evaluate the impact of BBR on browning of 3T3-L1. However, the oil drop can be stained in white adipocytes as well as brown or beige adipocytes. The expression of white adipocytes marker should be assessed in this condition.

Response:

A DM- or BBR treatment-mediated decrease in Resistin and BMP2 transcripts, a WAs-specific hormone and transcription factor, was shown in the revised manuscript (please see Figure 1b). Moreover, the result of Mitotracker Red FM staining was provided to demonstrate the induced mitochondriogenesis of 3T3-L1 cells with BBR treatment (please see Figure 1a).

4. In several quantitative results with statistical significance, the mark of significance ‘*’ are put in parentheses. The meaning of parentheses should be demonstrated.

Response:

The statistical significance was properly marked throughout the revised manuscript to clearly illustrate the results (please see Figures 1, 4, 5, 6, and 7).

5. In materials and methods section, the authors stated that the relative mRNA level subjected to qPCR was normalized to that of GAPDH. But there are also GAPDH level in qPCR data (figure 2b and c, figure 3c). Is it right that qPCR was performed instead of quantifying RT-PCR data?

Response:

It is right that the results of qPCR assays demonstrated the relative expressions of specific targets normalized to that of GAPDH. Figures 2 and 3 were revised according to the comment of reviewer.

6. In figure 5, authors used miR-92a-m3p and miR-92a-m5p as mutant miR-92a. It is necessary to explain what they are. Additionally, miR-92a-m3p did not have resistance as a mutant unlike miR-92a-m5p. The meaning of these results should be addressed.

Response:

The manuscript was revised according to the comment of reviewer (please see Results sec. 3.5, line 298-306).

Reviewer 4 Report

In the current manuscript, the authors describe that berberine induces beige adipogenesis increasing RBM4a expression by decreasing c13orf25 promoter-mediated miR-92a transcription. These serial response upon berberine promoted BA-specific splicing network, expression of beige cell marker genes, and finally respiratory rates of the cells. This observation is interesting but there are several issues with the work:

1. The introduction and discussion are somewhat insufficient and the overall justification for performing the study is poorly defined.

2. Relative mRNA level should be normalized to the level of GAPDH mRNA. This would present the relative values of mRNA level of other genes by setting the GAPDH level to 1. In this manuscript, all qPCR data presented different GAPDH levels and this is confusing how they organize the data.

3. In figure 3a, authors identified miR-92a-5p is relevant to BBR treatment through transcriptomic assays. Then, did miR-485 level not change in the assay results? According to Figure 3b, the miR-485 level is also likely to change.

4. While Oil red o staining data in figure 1a showed a great extent of accumulated oil drop, staining data in figure 6a and 7a showed very sparse oil drop is observed. What made this discrepancy?

5. In figure 6d, miR-92a-m3p exhibited the similar effects with miR-92a-5p on bioenergetic characteristics. Then, have the authors tested the impact of miR-92a-m3p in the level of brown adipocytes marker genes or BA-related splicing profiles?

6. In figure 6c and 7c, miR-92a-5p and shRBM4a increased PRDM16-ex16 in GM condition. Could the authors propose any explanation/speculation about it?

Author Response

Response to Reviewers' Comments

(Manuscript Number: Cells-521082)

Dear Editor:

We thank you for your response and for allowing revision of our manuscript (Cells-521082; Berberine promotes beige adipogenic signatures of 3T3-L1 cells by

regulating post-transcriptional events). The manuscript was revised in line with the valuable suggestions and comments of all reviewers. The revised manuscript was edited by a native English-speaking professional editor prior to submission. We hope that the revised manuscript achieves reviewer satisfaction. Our point-by-point responses to all specific reviewer comments, suggestions, and queries are as follows.

REVIEWER(S)' COMMENTS

Reviewer #4: In the current manuscript, the authors describe that berberine induces beige adipogenesis increasing RBM4a expression by decreasing c13orf25 promoter-mediated miR-92a transcription. These serial response upon berberine promoted BA-specific splicing network, expression of beige cell marker genes, and finally respiratory rates of the cells. This observation is interesting but there are several issues with the work:

Response to Reviewer Comments

1. The introduction and discussion are somewhat insufficient and the overall justification for performing the study is poorly defined.

Response:

The introduction and discussion was revised according to the comment of reviewer (please see Introduction and Discussion section).

2.  Relative mRNA level should be normalized to the level of GAPDH mRNA. This would present the relative values of mRNA level of other genes by setting the GAPDH level to 1. In this manuscript, all qPCR data presented different GAPDH levels and this is confusing how they organize the data.

Response:

The presentation of qPCR results was revised according to the comment of reviewer (please see Figure 2b, 2c, and 3c).

3.  In figure 3a, authors identified miR-92a-5p is relevant to BBR treatment through transcriptomic assays. Then, did miR-485 level not change in the assay results? According to Figure 3b, the miR-485 level is also likely to change.

Response:

The results of miRNA sequencing, RT-PCR and qPCR assays consistently demonstrated the impact of BBR treatment on enhancing the expression of miR-485 (please see Figure 3b, and 3c).

4. While Oil red o staining data in figure 1a showed a great extent of accumulated oil drop, staining data in figure 6a and 7a showed very sparse oil drop is observed. What made this discrepancy?

Response:

The accumulation of oil droplets in the BBR-treated 3T3-L1 cells was widely characterized on visual observation (Figure 1a), whereas the number or sized of lipid drop was different in individual cells on microscopic examination (Figure 6a and 7a). Nevertheless, BBR treatment exhibited a global influence on enhancing lipid accumulation in almost all treated cells.

5. In figure 6d, miR-92a-m3p exhibited the similar effects with miR-92a-5p on bioenergetic characteristics. Then, have the authors tested the impact of miR-92a-m3p in the level of brown adipocytes marker genes or BA-related splicing profiles?

Response:

Overexpression of miR-92a-m3p was demonstrated to exhibit similar effects to that observed in miR-92a-5p-overexpressing cells on changing the levels of brown adipocytes markers or BA-related splicing profiles.

6. In figure 6c and 7c, miR-92a-5p and shRBM4a increased PRDM16-ex16 in GM condition. Could the authors propose any explanation/speculation about it?

Response:

In non-differentiating cells, overexpression of miR-92a-5p or targeting of RBM4a reduced the relative levels of PRDM16-ex16 transcripts which were predominantly generated by mature brown adipocytes (Chi and Lin, Biochim. Biophys. Acta. Mol. Cell Res. 2018, 1865(11 Pt A), 1515-1525). This results demonstrated the effect of overexpressing miR-92a-5p on repressing BA-specific splicing event.

Round 2

Reviewer 3 Report

The responses to my concerns were well addressed.

Reviewer 4 Report

I think the present form is adequate for publishing.